# Spectral Dual-Layer Computed Tomography Can Predict the Invasiveness of Ground-Glass Nodules: A Diagnostic Model Combined with Thymidine Kinase-1

**DOI:** 10.3390/jcm12031107

**Published:** 2023-01-31

**Authors:** Tong Wang, Yong Yue, Zheng Fan, Zheng Jia, Xiuze Yu, Chen Liu, Yang Hou

**Affiliations:** 1Department of Radiology, Shengjing Hospital of China Medical University, Shenyang 110004, China; 2Department of Orthopedics, Shengjing Hospital of China Medical University, Shenyang 110004, China; 3Philips (China) Investment Co., Ltd., Shanghai 200072, China

**Keywords:** ground-glass nodule, lung adenocarcinoma, spectral computed tomography, quantitative and qualitative parameters, thymidine kinase-1

## Abstract

Objectives: Few studies have explored the use of spectral dual-layer detector-based computed tomography (SDCT) parameters, thymidine kinase-1 (TK1), and tumor abnormal protein (TAP) for the detection of ground-glass nodules (GGNs). Therefore, we aimed to evaluate the quantitative and qualitative parameters generated from SDCT for predicting the pathological subtypes of GGN-featured lung adenocarcinoma combined with TK1 and TAP. Material and Methods: Between July 2021 and September 2022, 238 patients with GGNs were retrospectively enrolled in this study. SDCT and tests for TK1 and TAP were performed preoperatively, and the lesions were divided into glandular precursor lesions (PGL), minimally invasive adenocarcinoma (MIA), and invasive adenocarcinoma (IAC), according to the pathological results. A receiver operating characteristic (ROC) curve was used to compare the diagnostic performance of these parameters. Multivariate logistic regression analysis was performed to construct a joint diagnostic model and create a nomogram. Results: This study included 238 GGNs, including 41 atypical adenomatous hyperplasias (AAH), 62 adenocarcinomas in situ (AIS), 49 MIA, and 86 IAC, with a high proportion of women, non-smokers, and pure ground-glass nodule (pGGN). CT100 keV (a/v), electronic density (EDW) (a/v), Daverage, Dsolid, TK1, and TAP of MIA and IAC were higher than those of PGL. The effective atomic number (Zeff (a/v)) was lower in MIA and IAC than in PGL (all *p* < 0.05). Logistic regression analysis showed that Zeff (a), EDW (a), TK1, Daverage, and internal bronchial morphology were crucial factors in predicting the aggressiveness of GGN. Zeff (a) had the highest diagnostic performance with an area under the ROC curve (AUC) = 0.896, followed by EDW (a) (AUC = 0.838) and CT100 keVa (AUC = 0.819). The diagnostic model and nomogram constructed using these five parameters (Zeff (a) + EDW (a) + CT100 keVa + Daverage + TK1) had an AUC = 0.933, which was higher than the individual parameters (*p* < 0.05). Conclusions: Multiple quantitative and functional parameters can be selected based on SDCT, especially Zeff (a) and EDW (a), which have high sensitivity and specificity for predicting GGNs’ invasiveness. Additionally, the combination of TK1 can further improve diagnostic performance, and using a nomogram is helpful for individualized predictions.

## 1. Introduction

Ground-glass nodules (GGNs) are a common manifestation, and studies have shown that persistent GGNs have a high malignancy rate and are predominant in early-stage lung adenocarcinoma [1]. Preoperative clarification of the relationship between radiological features and the degree of infiltration is essential for GGN management and appropriate surgical selection, thus reducing over-diagnosis or over-treatment. The detection rate of GGNs has increased with the gradual popularization of high-resolution computed tomography (HRCT) and low-dose computed tomography (LDCT) [2]. Previous research has focused on the morphological characteristics, size, solid components, and CT values of GGNs. Nevertheless, a meta-analysis concluded that a single radiological sign has limitations in discriminating pre-invasive and invasive adenocarcinoma, with a pooled sensitivity and specificity of 0.41~0.52 and 0.56~0.63, respectively [3].

Philips’ newly introduced IQon, a spectral dual-layer detector-based CT (SDCT), utilizes a dual-layer detector for high- and low-energy X-ray conversion and a stereoscopic data-acquisition system for parallel transmission. These capabilities enable simultaneous, isotropic, homologous, synchronous, and precise energy signal separation scans, providing a wider range of virtual single-energy images (MonoE) and more sets of parameters, such as an effective atomic number (Zeff) and electronic density (EDW) [4]. Compared with dual-energy CT, SDCT has a greater potential for noise reduction and optimizing image quality [5,6]. Moreover, a recent study confirmed that SDCT is feasible for identifying the aggressiveness of pure ground-glass nodules (pGGN) [7].

The concept of glandular precursor lesions (PGL) was proposed in the 2021 WHO classification [8], which included atypical adenomatous hyperplasia (AAH), adenocarcinomas in situ (AIS), and invasive adenocarcinoma (minimally invasive adenocarcinoma (MIA) and invasive adenocarcinoma (IAC)). PGL presents an inert growth behavior biologically, and it has a good prognosis. The progression from PGL and MIA to IAC is a continuous process accompanied by neoplastic cell proliferation and alignment changes. Thymidine kinase-1 (TK1) is a quantitative marker of cell proliferation that can be detected serologically; it is a key enzyme involved in the synthesis of DNA precursors [9]. TK1 has an elevated concentration in the serum of cancer patients, whereas it is extremely low or undetectable in healthy individuals or benign diseases (*p* < 0.0001) [10]. Tumor abnormal protein (TAP), a tumor marker, is highly expressed in precancerous lesions. Both TAP and TK1 contribute to the early detection of lung cancer and the screening of high-risk groups [11,12].

Few studies have explored the use of SDCT’s quantitative and qualitative parameters, TK1, and TAP for GGN assessment; therefore, this study aimed to investigate whether the aforementioned early-screening tools could effectively discriminate PGL, MIA and IAC and to establish a diagnostic model and nomogram.

## 2. Materials and Methods

### 2.1. Clinical Data

This retrospective study was approved by the hospital ethics committee (No. 2022PS1055K), and informed consent was obtained from all patients. Patients who underwent enhanced SDCT between July 2021 and September 2022 were included. The inclusion criteria were as follows: (1) single lesion with a maximum diameter of ≤30 mm (on lung window), with no involvement of lymph nodes or distant metastases; (2) preoperative TK1 and TAP tests; (3) postoperative pathology-confirmed AAH, AIS, MIA, or IAC; and (4) no history of preoperative adjuvant antitumor therapy. The exclusion criteria were as follows: (1) multiple GGNs, (2) incomplete clinical data or no surgical or pathological findings; (3) previous history of preoperative tumor treatment; and (4) nodules containing a cavity or vacuoles of diameter ≥5 mm. These nodules were excluded as the measurement and composition analysis of the spatiotemporal cavity can affect the results. Moreover, previous research has shown that the presence of the vacuole is of little value in the discrimination of GGN [13] (Figure 1).

### 2.2. SDCT Scan Technique

All patients underwent a three-phase chest enhanced scanning using SDCT (IQon Spectral CT, Philips Healthcare, Best, The Netherlands), wherein 50–80 mL of the contrast medium (Iodixanol, 270 mg/mL, GE Healthcare) was injected through the cubital vein, followed by 20–30 mL of normal saline at flow rates of 2.0–3.0 mL/s. The patients held their breath during the acquisition of the arterial and venous phase images 25 s and 60 s after the start of the injection, respectively.

The following acquisition parameters were used: tube current modulation, 120 kVp; rotation speed, 0.33 s/rot; helical pitch, 0.671; collimation, 64 × 0.625 mm; and matrix, 512 × 512. Recon mode iDose-level 3, filter standard (B) was reviewed on mediastinal windowing, and Y-Detail (YB) for lung windowing was applied to reconstruct the spectral base image with a slice thickness of 1 mm at 1 mm increments.

### 2.3. Image Analysis

Further image analysis was performed using a post-processing workstation (IntelliSpace Portal Version 6.5, Philips Healthcare). The region of interest (ROI) was selected semi-automatically (automatic recognition aided by manual modification) on the lung window (mGGN with the largest diameter containing the solid component) and synchronized to MonoE at 40 keV and 100 keV, iodine density map (IC), Zeff, and EDW. The ROI should be as large as possible, covering more than 80% of the lesion area while avoiding large bronchi, vessels, and air-containing cavities. The copy and paste function was used to ensure that the size and position of the ROI were the same between the arterial phase and venous phase.

All measurements were performed independently by two senior radiologists (with 8 and 10 years of experience in thoracic radiological diagnosis) under double-blind conditions, and the mean values were calculated. The following parameters were obtained: (1) CT value (HU), the mean value of the unenhanced phase under hybrid energy, CT40 keV, and CT100 keV (MonoE); (2) the slope of the spectral curve (λHU) = |CT40 keV − CT100 keV|/(100 − 40) (the large slope between 40 keV and 100 keV; higher energy levels than 100 keV had a relatively flat curve); (3) normalized iodine density map (NIC) = IC/ICaorta, where IC was normalized to the same level of thoracic aorta or subclavian artery to minimize the differences in patient hemodynamics and contrast dose distribution; and (4) enhancement difference value (EDV) = NICv-NICa/NICa.

### 2.4. TK1 and TAP Testing

TK1: On an empty stomach, 3 mL of peripheral venous blood was drawn and centrifuged at 3000 r/min for 10 min. The serum was separated and stored at −20 °C. An ELISA Kit (Shanghai Fuyu Biotechnology Co., Ltd., Shanghai, China) was used in strict accordance with the manufacturer’s instructions. The normal range was 0–2 pmol/L.

TAP: On an empty stomach, 1 mL of peripheral venous blood was collected for a blood smear. The coacervate and TAP detection systems (Shanghai Zhenke Biotechnology Co., Ltd.) were used in strict accordance with the manufacturer’s instructions. The reference values of the TAP agglutination area were as follows: <121 μm^2^, normal/no visible agglutination; 121 μm^2^ ≤ agglutination area <225 μm^2^, abnormal/low agglutination; and agglutination area ≥225 μm^2^, abnormal/large agglutination.

### 2.5. Statistical Analysis

SPSS (R26.0, IBM, Armonk, NY, USA) and MedCalc (Version 19.6.4, Ostend, Belgium) were used for statistical analyses. Continuous variables were expressed as mean ± standard deviation or median and interquartile P50 (P25, P75), respectively. The count data were expressed as (n, %). The Mann–Whitney U test and Kruskal–Wallis test were used for comparing data with non-normal distributions. Normally distributed data were compared using a *t*-test or a Fisher’s test. The count data were compared using chi-square tests. The intra-group correlation coefficient (ICC) was used to calculate the agreement between the assessments of the two readers. The receiver operating characteristic (ROC) curve was used to compare the diagnostic performance with the Youden index setting’s highest performance threshold. Univariate and multiple logistic regression analyses were used to construct a joint diagnostic model. Model calibration was evaluated using the Hosmer–Lemeshow test, and model discrimination was calculated by the Z test. A nomogram was constructed using R version 4.2.0. The statistical significance was set at *p* ≤ 0.05.

## 3. Results

A total of 238 GGNs met the inclusion criteria and were enrolled in this study. Women, non-smokers, and pGGN accounted for 58.40%, 67.23%, and 60.08%, respectively (*p* < 0.05). Ninety-one (38.24%) lesions were present in the RUL, and the postoperative pathological diagnosis finally confirmed 41 AAHs, 62 AISs (Figure 2), 49 MIAs, and 86 IACs (Figure 3) (Table 1). The interobserver agreement for the measurements by the readers was excellent (ICC = 0.81–0.92). Age, smoking history, and GGN nature differed significantly among PGL, MIA, and IAC (*p* < 0.05), as did smoking history and GGN nature in pairwise comparisons (Table 2).

Among the quantitative indicators, CT value, CT40 keV(a/v), CT100 keV(a/v), EDW (a/v), Daverage, Dsolid, TK1, and TAP of MIA and IAC were higher than those of PGL (all *p* < 0.05). Zeff (a/v) and EDV were lower in MIA and IAC than those in PGL (both *p* < 0.05). Further two-by-two comparisons showed statistical differences between CT100 keV (a/v), Zeff (a/v), and EDW (a/v) (*p* < 0.05). Among the morphological signs, the differences between the three groups in terms of margin, internal vascular morphology, internal bronchial morphology, and pleural indentation were statistically significant (*p* < 0.05), and only internal bronchial morphology differed significantly in pairwise comparisons (*p* < 0.05).

Univariate and multifactorial logistic regression analyses revealed that Zeff (a), EDW (a), TK1, Daverage, and internal bronchial morphology were significant factors for predicting GGN aggressiveness (*p* < 0.05) (Table 3). Spearman’s correlation analysis revealed that Zeff (a/v) was negatively correlated with invasiveness (r = −0.699/−0.589, *p* < 0.001), and EDW (a/v), CT value, CT100 keV (a/v), TK1, and Daverage were positively correlated with invasiveness, with r of 0.633/0.625, 0.6, 0.59/0.552, 0.398, and 0.484, respectively; (*p* < 0.001). The diagnostic efficacy and optimal thresholds for the abovementioned parameters were analyzed (Table 4). Among them, Zeff (a) showed the highest diagnostic performance with AUC = 0.896, sensitivity = 88.15%, and specificity = 78.64%, followed by EDW (a) (AUC = 0.838, sensitivity = 66.67%, and specificity = 89.32%), CT100 keVa (AUC = 0.819), and CT value (AUC = 0.816). TK1 (AUC = 0.733), and Daverage (AUC = 0.739) also showed moderate diagnostic values.

Quantitative parameters with high diagnostic efficacy were selected to construct diagnostic model 1 and the ROC curve shown below (Table 5) with AUC = 0.919 (0.877–0.950) > 0.75, which demonstrated good discrimination and calibration (Hosmer–Lemeshow χ^2^ = 8.270, *p* = 0.408 > 0.05). When TK1 was incorporated to obtain diagnostic model 2, the overall diagnostic efficacy and specificity improved, with AUC = 0.933 (0.894–0.961) and Hosmer–Lemeshow χ^2^ = 2.746, and *p* = 0.949, indicating that model 2 also had better discrimination and calibration. Both models had higher diagnostic efficacy than the individual parameters (*p* < 0.05) (Figure 4). These five parameters were used to build a nomogram (Figure 5), with each feature corresponding to the value of the score in the uppermost scale and the sum of the scores corresponding to the hazard coefficient on the lowermost axis.

## 4. Discussion

In this study, model 2 constructed with five parameters (Zeff (a) + EDW (a) + CT100 keVa + Daverage + TK1) had a good ability to discriminate PGL from adenocarcinoma (AUC = 0.933). The efficacy of this diagnostic model was higher than model 1 or individual parameters (*p* < 0.05), providing a new method for the noninvasive identification of GGN and reducing subjective bias. Furthermore, the quantitative risk of invasiveness of a GGN can be accurately calculated using the nomogram.

An appropriate differentiation between PGL, MIA, and IAC is crucial for the selection of surgical approaches and prognosis. Our results illustrated that Zeff (a) was higher in PGL than in adenocarcinoma, correlated negatively with infiltration, showed a unique advantage (AUC = 0.896) with a threshold of ≤9.04, and possessed high sensitivity (88.15%) and specificity (78.64%), indicating that Zeff can monitor the changes in tumor cell appendage growth components and structure during the development of early-stage adenocarcinoma.

Additionally, EDW (a) also had a high diagnostic performance (AUC = 0.838), was an independent predictor of GGN aggressiveness, and positively correlated with the degree of invasion, which conformed to Zhang et al.’s results [14]. There are currently few examples of research on Zeff and EDW for predicting GGN. A study by Yu et al. showed that the ED-Zeff ratio in the plain phase was an independent predictor of IA, whereas our results slightly differ from Yu et al.’s viewpoint, as they aimed to differentiate MIA from IA manifesting as pGGNs [7].

Zeff assigns material component information to each pixel, creating a colorful image that visualizes peri-tumor boundaries [15] and facilitates the detection of shallow, tiny GGNs, especially under the interference of uneven lung permeability or pneumonia. Increased malignant cells in invasive adenocarcinomas cause a progressive addition in the lipid composition and water content of the lymphatic vessels, whereupon a decrease in Zeff was induced [7]. Further, EDW shows a relative distribution plot of electron density corresponding to each voxel without requiring conversion to CT values, and the results are more accurate [16]. The advancement of PGL-MIA-IAC was accompanied by increased numbers of deteriorating tumor cells, resulting in the thickening of the alveolar cavity, the collapse of the alveoli, and decreased intraluminal gas, with the appearance of elevated EDW [14].

As the percentage of lepidic growth components in GGN gradually decreases and the density increases, CT100 keV (Threshold > −458.6) and CT value (Threshold > −495.2) had similar diagnostic efficacy and specificity with good performance, which was similar to the results of the studies by Zhan et al. [17] and Yu et al. [18]. This study also demonstrated that CT100 keV improved the signal-to-noise ratio and contrast to optimize the visibility of GGN compared with CT40 keV.

Meanwhile, TK1 was superior to TAP in identifying GGN aggressiveness. The positive correlation between the aggressiveness of TK1 and GGN in our cohort confirmed that TK1 was a reliable biomarker for evaluating precancerous cells, superior to carcinoembryonic antigens [11,12]. A single biomarker has difficulty in meeting the clinical requirements when TK1 alone with moderate discriminatory efficacy (AUC = 0.733) is applied. When combining the quantitative parameters of SDCT with TK1 for diagnosis, we found that model 2 could improve the overall accuracy and specificity to some extent.

Classical CT parameters are associated with GGN extension or invasion [19,20]. In this study, we found that the proportions of Daverage, Dsolid, and internal bronchial morphology were higher in adenocarcinoma than those in PGL, as in the previous studies [21,22]. We observed no significant difference between λHU and NIC in PGL, MIA, and IAC; the diagnostic efficacy was low, which is consistent with Wang et al. [13] and Zhang et al. [23]. The proliferation of immature neovascularization accompanies the growth from PGL to adenocarcinoma, but the vascular variation is a histological transition, and it is difficult for λHU and NIC to correctly recognize this complex transition.

## 5. Limitations

This study had some limitations. First, this was a single-center, retrospective study with patient-selection bias coupled with a limited number of cases; therefore, further investigation with a larger sample size is needed. Second, we combined quantitative SDCT parameters, some morphological features, TKl, and TAP to assess GGN. In the future, we will further compare radiomics and add more preoperative diagnostic information. Third, we did not outline the foci in all three dimensions, which relied on a functional upgrade of the post-processing workstations.

## 6. Conclusions

In conclusion, multiple quantitative and functional parameters can be selected based on SDCT, especially Zeff (a) and EDW (a), which have high sensitivity and specificity for predicting the pathological subtypes and risk stratification of GGNs. In addition, the combination of TK1 can further improve diagnostic performance. Using a nomogram is helpful for individualized predictions.

## Figures and Tables

**Figure 1 jcm-12-01107-f001:**
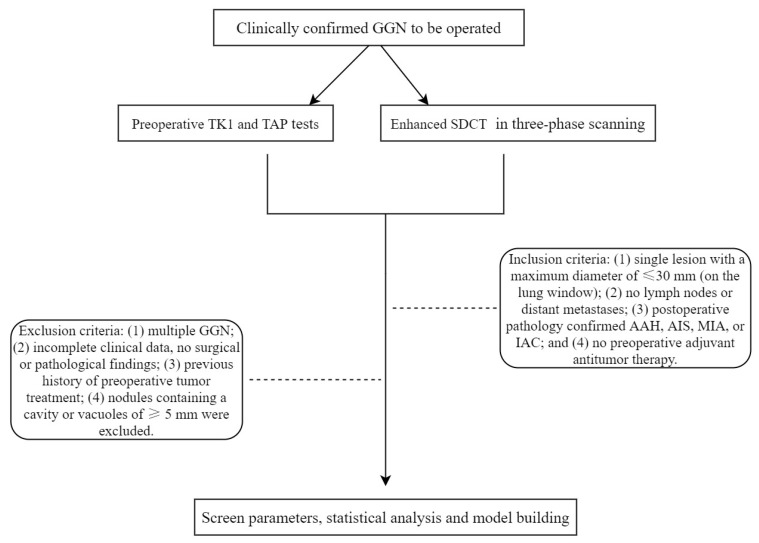
Study flow chart. GGN, ground-glass nodule; TK1, thymidine kinase-1; TAP, tumor abnormal protein; SDCT, spectral dual-layer detector-based computed tomography; AAH, atypical adenomatous hyperplasia; AIS, adenocarcinoma in situ; MIA, minimally invasive adenocarcinoma; IAC, invasive adenocarcinoma.

**Figure 2 jcm-12-01107-f002:**
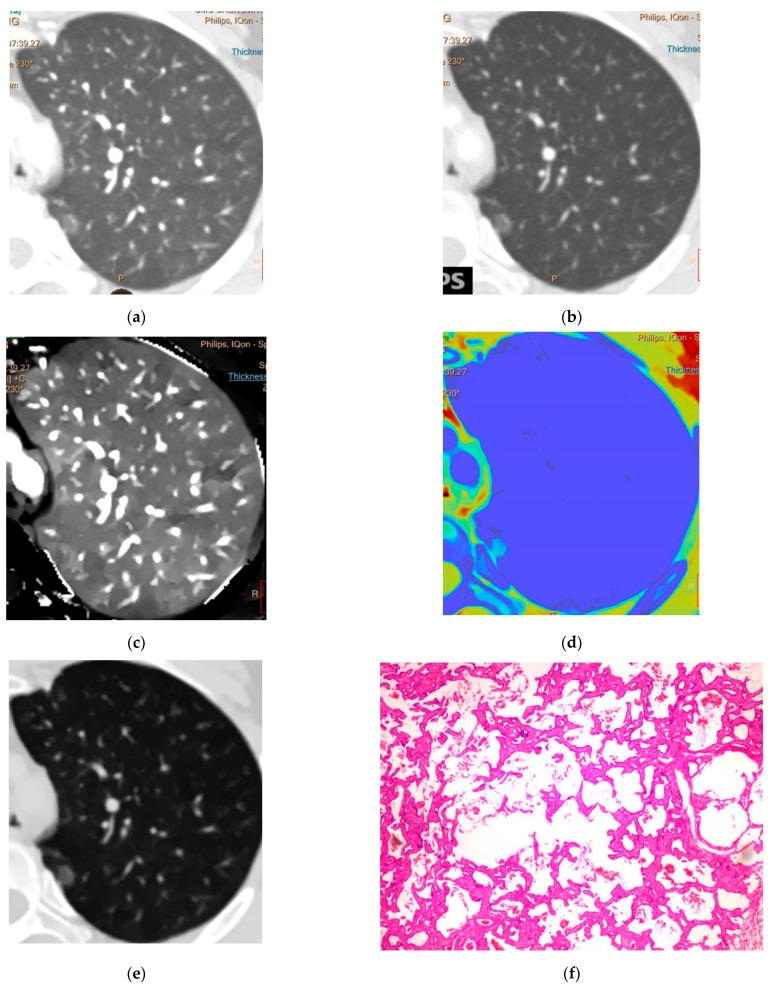
(**a**) CT40 keV, (**b**) CT100 keV, (**c**) iodine density map, (**d**) effective atomic number, and (**e**) electron density in the arterial phase. Spectral dual-layer detector-based computed tomography of a 43-year-old woman showing pGGN in the left upper lobe of lung, Daverage = 8.95 mm, CT40 keVa = −410.5, CT100 keVa = −548.7, IC (a) = 1.86, Zeff (a) = 9.37, and EDW (a) = 41.2. The postoperative pathology is adenocarcinoma in situ (**f**).

**Figure 3 jcm-12-01107-f003:**
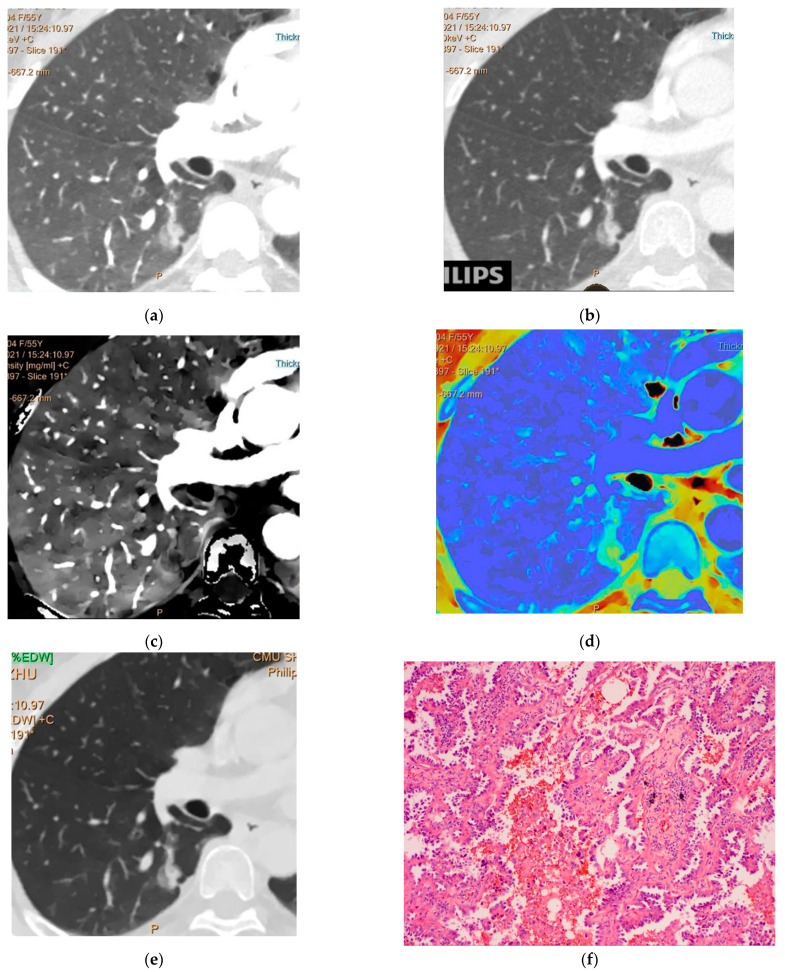
(**a**) CT40 keV, (**b**) CT100 keV, (**c**) iodine density map, (**d**) effective atomic number, and (**e**) electron density in the arterial phase. Spectral dual-layer detector-based computed tomography of a 55-year-old woman showing mGGN in the lower lobe of right lung, Daverage = 14.78 mm, CT40 keVa = −141.7, CT100 keVa = −223.8, IC (a) = 2.11, Zeff (a) = 8.19, and EDW (a) = 77.3. The postoperative pathology is invasive adenocarcinoma (**f**).

**Figure 4 jcm-12-01107-f004:**
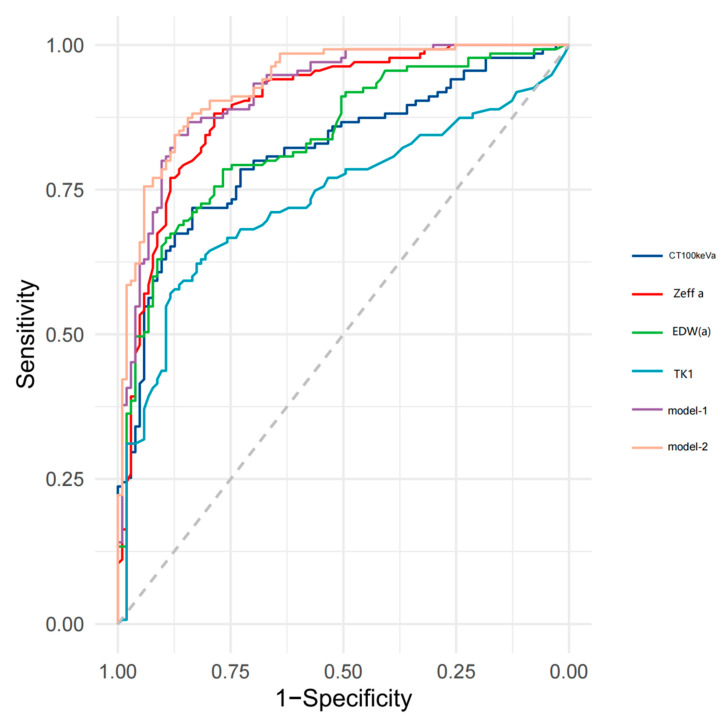
Comparison of the efficacy of the joint diagnostic models 1 and 2 with spectral dual-layer detector-based computed tomography (SDCT)-derived quantitative parameters. Zeff, effective atomic number; EDW, electron density; TK1, thymidine kinase-1; model-1, Zeff (a) + EDW (a) + CT100 keVa + Daverage; model-2, Zeff (a) + EDW (a) + CT100 keVa + Daverage + TK1; AP, arterial phase; VP, venous phase.

**Figure 5 jcm-12-01107-f005:**
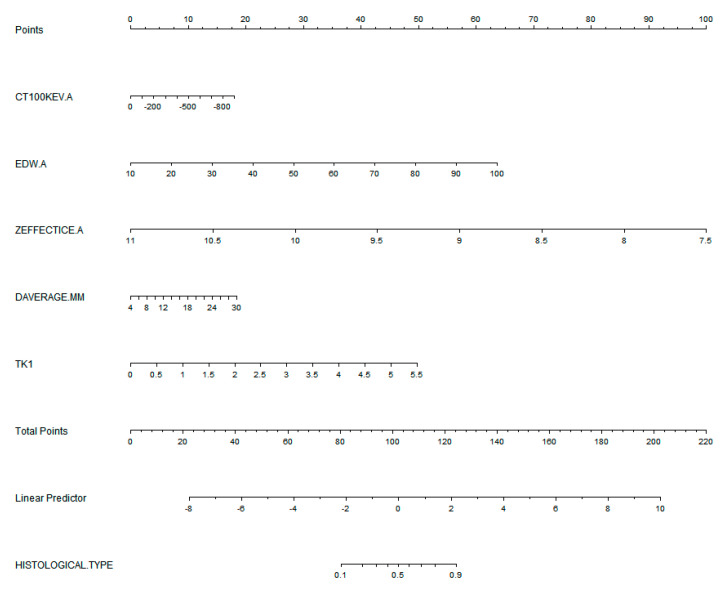
Nomogram created by combining the parameters predicting ground-glass nodules (GGN) featured lung adenocarcinoma invasiveness. Zeff, effective atomic number; EDW, electron density; TK1, thymidine kinase-1.

**Table 1 jcm-12-01107-t001:** Clinical information and histologic results of the patients.

Characteristics	Values (n = 238)
Age, range (median)	24–79 (58)
Sex, n (%)	
Female	139 (58.40%)
Male	99 (41.60%)
Smoking, n (%)	
Non-smoker	147 (67.23%)
Smoker	35 (32.77%)
GGN type, n (%)	
pGGN	143 (60.08%)
mGGN	95 (39.92%)
GGN location, n (%)	
RUL	91 (38.24%)
RML	12 (5.04%)
RLL	41 (17.23%)
LUL	57 (23.95%)
LLL	37 (15.55%)
Histological types, n (%)	
AAH	41 (17.23%)
AIS	62 (26.05%)
MIA	49 (20.59%)
IAC	86 (36.13%)

pGGN, pure ground-glass nodule; mGGN, mixed ground-glass nodule; RUL, right upper lobe; RML, right middle lobe; RLL, right lower lobe; LUL, left upper lobe; LLL, left lower lobe; AAH, atypical adenomatous hyperplasia; AIS, adenocarcinoma in situ; MIA, minimally invasive adenocarcinoma; IAC, invasive adenocarcinoma.

**Table 2 jcm-12-01107-t002:** Comparison between the quantitative parameters of GGN-featured lung adenocarcinoma of precursor glandular lesions and adenocarcinoma.

Parameter		Precursor Glandular Lesions (n = 103)	Adenocarcinoma (n = 135)		*p* Value			
		AAH (n = 41) + AIS (n = 62)	MIA (n = 49)	IAC (n = 86)		Pa	Pb	Pc
Sex	female	54	30	55	0.251	0.31	0.111	0.753
	male	49	19	31				
Age		55.03 ± 11.03	58.41 ± 10.95	60.23 ± 9.44	0.003	0.064	<0.001	0.331
Smoking status	non-smoker	57	43	60	<0.001	<0.001	0.043	0.019
	smoker	46	6	26				
pGGN		85	28	30	<0.001	<0.001	<0.001	0.012
mGGN		18	21	56	<0.001	<0.001	<0.001	0.012
CT value		−550.00 ± 127.19	−427.00 ± 151.71	−257.55 (−408.50–121.20)	<0.001	<0.001	<0.001	<0.001
CT40 keVa		−440.20 (−530.70–370.45)	−328.37 ± 170.78	−191.26 ± 214.39	<0.001	<0.001	<0.001	<0.001
CT100 keVa		−583.80 (−656.25–508.42)	−453.23 ± 151.20	−310.05 (−436.40–149.30)	<0.001	<0.001	<0.001	<0.001
λHUa		2.07 (1.62–2.69)	2.08 ± 0.75	2.06 (1.42–2.54)	0.641	0.463	0.409	0.965
NICa		0.16 ± 0.06	0.17 ± 0.06	0.19 ± 0.07	0.029	0.262	0.007	0.257
Zeff (a)		9.52 ± 0.59	8.80 ± 0.47	8.49 ± 0.35	<0.001	<0.001	<0.001	<0.001
EDW (a)		36.89 ± 11.10	47.46 ± 12.94	62.36 ± 17.38	<0.001	<0.001	<0.001	<0.001
CT40 keVv		−478.70 (−563.00–401.20)	−349.56 ± 172.97	−201.45 (−344.70–36.40)	<0.001	0.003	<0.001	<0.001
CT100 keVv		−569.2 (−643.07–505.00)	−454.89 ± 154.69	−326.06 ± 186.29	<0.001	<0.001	<0.001	<0.001
λHUv		1.62 (1.25, 2.15)	1.76 ± 0.62	1.81 ± 0.76	0.469	0.586	0.227	0.602
NICv		0.32 ± 0.13	0.31 ± 0.09	0.33 ± 0.11	0.371	0.842	0.252	0.189
Zeff (v)		9.17 ± 0.48	8.72 ± 0.46	8.46 ± 0.36	<0.001	<0.001	<0.001	<0.001
EDW (v)		36.70 (30.35, 44.60)	49.26 ± 13.38	64.21 ± 17.45	<0.001	<0.001	<0.001	<0.001
Enhancement difference value (EDV)		1.27 ± 1.08	0.97 ± 0.88	0.88 ± 0.59	0.032	0.316	0.201	0.994
Daverage (mm)		10.00 (8.07, 13.27)	11.40 (9.38, 14.71)	17.96 ± 6.17	<0.001	0.034	<0.001	<0.001
Dsolid (mm)		0.57 ± 0.15	3.67 ± 2.09	5.05 ± 0.56	<0.001	0.006	<0.001	0.004
Margin	Spiculated/lobulated	41	18	58	<0.001	0.464	<0.001	0.002
Internal vascular morphology	Distorted/dilated/cut off	24	15	48	<0.001	0.119	<0.001	0.003
Internal bronchial morphology	Distorted/thickened/stiff	11	13	38	<0.001	<0.001	<0.001	0.024
Pleural indentation	Present	19	15	44	<0.001	0.062	<0.001	0.054
TK1		0.27 (0.14, 0.51)	0.80 (0.22, 1.98)	1.04 (0.34, 1.91)	<0.001	<0.001	<0.001	0.359
TAP		103.72 ± 26.99	120.76 ± 30.23	126.62 (101.09, 139.40)	<0.001	<0.001	<0.001	0.934

Pa for precursor glandular lesions vs. MIA; Pb for precursor glandular lesions vs. IAC; and Pc for MIA vs. IAC. AAH, atypical adenomatous hyperplasia; AIS, adenocarcinoma in situ; MIA, minimally invasive adenocarcinoma; IAC, invasive adenocarcinoma; pGGN, pure ground-glass nodule; mGGN, mixed ground-glass nodule; Zeff, effective atomic number; EDW, electron density; NIC, normalized iodine density map; λHU, slope of spectral curve; TK1, thymidine kinase-1; TAP, tumor abnormal protein; AP, arterial phase; VP, venous phase.

**Table 3 jcm-12-01107-t003:** Multifactorial analysis of GGN invasiveness.

Parameters	Estimate	Std. Error	Wald	*p* Value
Daverage	0.115	0.04	8.117	0.004
Dsolid	0.014	0.071	0.041	0.840
CT100 keV (a)	0.000	0.004	0.015	0.902
λHU (a)	−0.042	0.164	0.065	0.799
NIC (a)	−3.953	5.157	0.588	0.443
Zeff (a)	−2.458	0.534	21.184	<0.001
EDW (a)	0.977	0.513	3.619	0.046
CT100 kev (v)	−0.002	0.004	0.292	0.589
λHU (v)	0.011	0.255	0.002	0.966
NIC (v)	−1.364	2.989	0.208	0.648
Zeff (v)	0.427	0.589	0.526	0.468
Enhancement difference value (EDV)	−0.746	0.393	3.597	0.058
EDW (v)	0.033	0.072	0.211	0.646
TAP	0.007	0.006	1.223	0.269
TK1	0.606	0.185	10.708	0.001
Margin	−0.230	0.499	0.213	0.645
Internal vascular morphology	0.210	0.489	0.184	0.668
Internal bronchial morphology	−1.142	0.505	5.106	0.024
Pleural indentation	0.077	0.073	1.114	0.057

CT value and CT40 keV were not included because of multicollinearity. Zeff, effective atomic number; EDW, electron density; NIC, normalized iodine density map; λHU, slope of spectral curve; TK1, thymidine kinase-1; TAP, tumor abnormal protein; AP, arterial phase; VP, venous phase.

**Table 4 jcm-12-01107-t004:** The diagnostic efficiency of the parameters for distinguishing between precursor glandular lesions and adenocarcinoma.

Parameters	AUC (95%CI)	Youden Index	Threshold	Sensitivity (%)	Specificity (%)
CT Value	0.816 (0.760–0.863)	0.553	>−495.2	71.85	83.5
CT40 keVa	0.769 (0.710–0.821)	0.492	>−367.6	72.59	76.7
CT100 keVa	0.819 (0.764–0.865)	0.568	>−458.6	73.33	83.5
λHUa	0.536 (0.470–0.600)	0.096	≤1.49	25.19	84.47
NICa	0.592 (0.527–0.655)	0.184	>0.21	31.11	87.38
Zeff (a)	0.896 (0.850–0.932)	0.667	≤9.04	88.15	78.64
EDW (a)	0.838 (0.785–0.882)	0.559	>47.6	66.67	89.32
CT40 keVv	0.779 (0.721–0.830)	0.503	>−404.8	75.56	74.76
CT100 keVv	0.795 (0.738–0.844)	0.543	>−482.5	71.85	82.52
λHUv	0.542 (0.477–0.607)	0.159	>1.63	64.44	51.46
NICv	0.528 (0.462–0.593)	0.133	>0.31	57.04	56.31
Zeff (v)	0.833 (0.779–0.878)	0.518	≤8.87	78.36	73.53
EDW (v)	0.833 (0.779–0.878)	0.553	>45.9	71.85	83.5
Enhancement difference value (EDV)	0.553 (0.487–0.617)	0.125	≤1.83	94.07	18.45
Daverage	0.739 (0.678–0.793)	0.386	>11.05	75.56	63.11
Dsolid	0.691 (0.628–0.749)	0.376	>3.86	44.44	93.2
Margin	0.629 (0.564–0.690)	0.257	Spiculated/lobulated	70.37	55.34
Internal vascular morphology	0.648 (0.584–0.709)	0.296	Distorted/dilated/cut off	57.78	71.84
Internal bronchial morphology	0.685 (0.622–0.744)	0.370	Distorted/thickened/stiff	49.63	87.38
Pleural_indentation	0.651 (0.587–0.712)	0.302	Present	52.59	77.67
TK1	0.733 (0.672–0.788)	0.453	>0.87	57.04	88.35
TAP	0.679 (0.616–0.738)	0.303	>103.21	74.07	56.31

Zeff, effective atomic number; EDW, electron density; NIC, normalized iodine density map; λHU, slope of spectral curve; TK1, thymidine kinase-1; TAP, tumor abnormal protein; AP, arterial phase; VP, venous phase; AUC, area under the curve.

**Table 5 jcm-12-01107-t005:** Comparison of the diagnostic efficiency of the diagnostic models based on SDCT parameters and TK1.

	AUC (95%CI)	Sensitivity (%)	Specificity (%)	Hosmer–Lemeshow		
				χ^2^	*p* Value	
Model 1	0.919 (0.877–0.950)	91.110	87.38	8.270	0.408	Z = 2.542, *p* = 0.011
Model 2	0.933 (0.894–0.961)	88.890	88.89	2.746	0.949	

Model 1: Zeff a + EDW (a) + CT100 keVa + Daverage; Model 2: Zeff a + EDW (a) + CT100 keVa + Daverage + TK1; SDCT, spectral dual-layer detector-based computed tomography; Zeff, effective atomic number; EDW, electron density; TK1, thymidine kinase-1; AP, arterial phase; AUC, area under the curve.

## Data Availability

Data available on request due to restrictions on privacy. The data presented in this study are available on request from the corresponding author.

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
