# Peer review of "Spectral Dual-Layer Computed Tomography Can Predict the Invasiveness of Ground-Glass Nodules: A Diagnostic Model Combined with Thymidine Kinase-1"

_jcm, 2023, doi:10.3390/jcm12031107_

Round 1

Reviewer 1 Report

The Authors performed a nice retrospective study addressing the capability of a new CT technology to investigate aggressiveness of lesions as GGO which are established as adenocarcinoma precursors.

I found the paper of interest and well conducted.

Minor flaws

1. The Authors should not confuse clinical cases, as the one they present, and in which advanced CT with contrast media administration and substantial radiation dose exposures are justified with the screening of LC that entails no contrast media administration and low radiation dose technology.

2. The Authors should strive to better explain how the two most discriminant features extracted from SDSCT, namely Zeff and EDW and their changes from pre-invasive lesions to invasive AdenoK, are linked to cancer progression. This can be perceived between the lines, but a new dedicated paragraph would considerably help the readers.

3. The Authors had the opportunity to consider and insert in the model n.2 the results of the quantification of two biomarkers, namely Tk1 and TAP.  While this is per se a valuable addendum, certainly does not represent the main contribute of the paper and probably should be treated as such without too much emphasis.

English requires some minor revision.

Author Response

Response to Reviewer 1 Comments  

Reply to Reviewer #1

We feel great thanks for your professional review work on our article. As you are concerned, there are several problems that need to be addressed. According to your nice suggestions, we have made extensive corrections to our previous draft, the detailed corrections are listed below.In this revised version, changes to our manuscript were all highlighted within the document by using red-colored text.

1.Response to comment Point 1:(1. The Authors should not confuse clinical cases, as the one they present, and in which advanced CT with contrast media administration and substantial radiation dose exposures are justified with the screening of LC that entails no contrast media administration and low radiation dose technology.):

Reply: 

Thanks for your careful checks.I'm sorry that this had caused ambiguity.According to your nice suggestions, we have revised the manuscript(introduction section, (page 2 , line 49-57)) accordingly and highlighted in red. As you suggest, we should not confuse the clinical cases, conventional CT (HRCT and LDCT), which are frequently used to screen LC risk populations, has the advantages of low radiation dose and avoidance of contrast agents, and new CT techniques developed on this basis, while more valuable parameters have been excavated, inevitably increasing the radiation dose and use of contrast agents.

  1. Response to comment( The Authors should strive to better explain how the two most discriminant features extracted from SDSCT, namely Zeff and EDW and their changes from pre-invasive lesions to invasive AdenoK, are linked to cancer progression. This can be perceived between the lines, but a new dedicated paragraph would considerably help the readers.):

Reply: 

We feel great thanks for your professional review work on our article. According to your nice suggestions, We focus on explaining the two most valuable parameters (Zeff and EDW) extracted from SDCT and their correlation with cancer progression. These new contents form a new paragraph and are embedded in the discussion section (page 14 , line 82-92).

  1. Response to comment( The Authors had the opportunity to consider and insert in the model n.2 the results of the quantification of two biomarkers, namely Tk1 and TAP.  While this is per sea valuable addendum, certainly does not represent the main contribute of the paper and probably should be treated as such without too much emphasis.):

Reply: 

Thank you for your helpful suggestions and guidance.According to suggestion,We reduced the discussion of TK1 and TAP (page 14, line 106-114 ), as biomarkers,they are also valuable as a screening tool for lung cancer. We found that TK1 was more effective than TAP in identifying pre-invasive lesion and invasive adenocarcinoma, but less effective than SDCT-derived quantitative parameters(especially Zeff and EDW). Moreover, when TK1 inserted in the diagnostic model-2 and nomogram , it is helpful for individualized predictions.

  1. Response to comment(English requires some minor revision):

Reply: 

Thanks for your careful checks.I'm sorry that this had caused ambiguity. We also employed an English-Language editing service by MDPI to polish our wording.                We hope that the revision is acceptable.

Finally, we would like to thank the editors and reviewers again for the careful reading of our paper. We tried our best to improve and revise the manuscript carefully and believe that the new version is much better than the old one. We appreciate for Reviewers’ warm work earnestly, and hope the correction will meet with approval. Once again, thank you very much for your comments and suggestions.Hope the revised version is acceptable.

Thank you again for your comments and suggestions.

Best wishes,

Reviewer 2 Report

Comment 1: The title is a very lengthy and narrow spectrum, authors must review the title

Comment 2: The authors should define the abbreviations for the first time in the abstract, text of the manuscript, and figure legends, and then follow with the abbreviations in the whole of the manuscript.

Comments 3: In my opinion, the introduction section needs more clarity in objectives, Authors must explain clinically significant and insignificant adenocarcinoma, and also explain why and how the diagnostic model combined with thymidine kinase-1 predicts more accurately than other models. The authors need to improve it.

Comments 4; Please go over your manuscript text and ensure it is written in an acceptable English language.

Comments 5: The authors must describe a clear explanation of the suitable technique for prostate tumor characterization and introduction.

Comments 6: Are there uniformly accepted diagnostic criteria regarding the treatment of PCa tumors?

Comments 7: manuscript lack of latest developmental strategies of combined diagnostic combination for lungs tumor characterization and diagnosis

Comments 8: The manuscript lack of detailed explanation of Factors leading to the failure/limitations of the older version of diagnostics approaches for adenocarcinoma

Comments 9: It is suggested to check the manuscript for English grammar once more. Also, the authors have written very short sentences or non-academic sentences in the manuscript; therefore, they need to correct and edit with professional English editing.

Author Response

Response to Reviewer 2 Comments  

Reply to Reviewer #2

We feel great thanks for your professional review work on our article. As you are concerned, there are several problems that need to be addressed. According to your nice suggestions, we have made extensive corrections to our previous draft, the detailed corrections are listed below.In this revised version, changes to our manuscript were all highlighted within the document by using red-colored text.

1.Response to comment :(Comment 1: The title is a very lengthy and narrow spectrum, authors must review the title):

Reply: 

Thanks for your careful checks.I'm sorry that this had caused ambiguity.According to your nice suggestions, we have changed the title to make it more concise and beneficial to understand (page 1 , line 2-5).

  1. Response to comment(Comment 2:The authors should define the abbreviations for the first time in the abstract, text of the manuscript, and figure legends, and then follow with the abbreviations in the whole of the manuscript.):

Reply: 

We feel great thanks for your professional review work on our article. According to your nice suggestions, We check and define the abbreviations for the first time in the abstract, text of the manuscript, and figure legends.

  1. Response to comment(Comments 3:In my opinion, the introduction section needs more clarity in objectives, Authors must explain clinically significant and insignificant adenocarcinoma, and also explain why and how the diagnostic model combined with thymidine kinase-1 predicts more accurately than other models. The authors need to improve it.):

Reply: 

Thank you for your helpful suggestions and guidance.According to suggestion,We have improved the content of the introduction (page 2, line 46-49,72-75), explain the purpose and clinical significance of adenocarcinoma, to make it clearer. On the other hand, we have also revised the content of the discussion (page 14, line 111-116), strengthening the interpretation of why the diagnostic model-2 incorporating TK1 predicted more accurately than other model.

TK1, as biomarkers,is also valuable as a screening tool for lung cancer. We found that TK1 was more effective than TAP in identifying pre-invasive lesion and invasive adenocarcinoma, but less effective than SDCT-derived quantitative parameters(especially Zeff and ED). Moreover, when TK1 inserted in the diagnostic model-2 and nomogram , it is helpful for individualized predictions.

  1. Response to comment(Comments 4;Please go over your manuscript text and ensure it is written in an acceptable English language.):

Reply: 

Thanks for your careful checks.I'm sorry that this had caused ambiguity. We also employed an English-Language editing service to polish our wording.                We hope that the revision is acceptable.

  1. Response to comment(Comments 5:The authors must describe a clear explanation of the suitable technique for prostate tumor characterization and introduction.):

Reply: 

Thank you for your helpful suggestions and guidance. Conventional CT (HRCT and LDCT), which are frequently used to screen lung adenocarcinoma risk populations, has the advantages of low radiation dose and avoidance of contrast agents.Several studies have focused on the morphological features, size, solid components, and CT values of GGN,and new CT techniques(such as spectral dual-layer detector-based CT) developed on this basis, while more valuable parameters have been excavated.

PET(CT/MR) is less used in the evaluation of GGN, considering the high price and low radioactive concentration, with lower SUVmax. PET(CT/MR) is beneficial for the evaluation of advanced tumors, lymph node metastases or other metastases, etc.

  1. Response to comment(Comments 6:Are there uniformly accepted diagnostic criteria regarding the treatment of PCa tumors?):

Reply: 

Thank you for your helpful suggestions and guidance. Through consulting the literature, we learned that Prostate cancer (PCa) is the second malignant tumor in the world with the second incidence rate. In recent years, the incidence rate of PCa in China is on the rise. The diagnosis and treatment of PCa has formed a relatively unified consensus. At present, the relatively recognized early clinical diagnosis mode of PCa is the "three steps" method. In terms of treatment, early PCa can achieve good treatment effect through radical surgery or radical radiotherapy, etc., while metastatic prostate cancer still takes androgen deprivation therapy (ADT) as the first-line treatment plan. With the update of evidence-based evidence, the guidelines have been updated and revised, and new chapters added.

  1. Response to comment(Comments 7:manuscript lack of latest developmental strategies of combined diagnostic combination for lungs tumor characterization and diagnosis):

Reply: 

Thank you for your helpful suggestions and guidance.According to suggestion,We have improved the content of the introduction and discussion(page 2 , line 49-53, line 61-64, line 78-84) (page 14 , line 82-92, line 115-117) about latest development strategy of lung tumor combined diagnosis. Early screening with low dose CT can significantly reduce the risk of death from lung cancer. However, the high false positive rate of CT screening brings some difficulties for follow-up diagnosis and treatment.With new CT techniques developed, more valuable parameters have been excavated.Serological tumor markers are also a common clinical means to assist diagnosis.In recent years, liquid biopsy (such as ctDNA, miRNA, tumor specific antibodies, etc.) has become possible for screening and diagnosis of early lung adenocarcinoma. Reasonable application of AI will be conducive to further improving the clinical benefits. And that's what we're going to do.

  1. Response to comment(Comments 8:The manuscript lack of detailed explanation of Factors leading to the failure/limitations of the older version of diagnostics approaches for adenocarcinoma):

Reply: 

Thank you for your helpful suggestions and guidance.According to suggestion,We have improved the content of the introduction and discussion (page 14, line 111-116),explain the reasons for the limitations of early diagnosis of lung adenocarcinoma in diagnostics approaches, to make it clearer.

  1. Response to comment(Comments 9:It is suggested to check the manuscript for English grammar once more. Also, the authors have written very short sentences or non-academic sentences in the manuscript; therefore, they need to correct and edit with professional English editing.):

Reply: 

Thanks for your careful checks.I'm sorry that this had caused ambiguity. We carefully revised the English version of the manuscript to make the language smooth, and also employed an English-Language editing service to polish our wording.

Finally, we would like to thank the editors and reviewers again for the careful reading of our paper. We tried our best to improve and revise the manuscript carefully and believe that the new version is much better than the old one. We appreciate for Reviewers’ warm work earnestly, and hope the correction will meet with approval. Once again, thank you very much for your comments and suggestions.Hope the revised version is acceptable.

Thank you again for your comments and suggestions.

Best wishes,
